# N-Fertilizer (Urea) Enhances the Phytoextraction of Cadmium through *Solanum nigrum* L.

**DOI:** 10.3390/ijerph17113850

**Published:** 2020-05-29

**Authors:** Arosha Maqbool, Shafaqat Ali, Muhammad Rizwan, Muhammad Saleem Arif, Tahira Yasmeen, Muhammad Riaz, Afzal Hussain, Shamaila Noreen, Mohamed M. Abdel-Daim, Saad Alkahtani

**Affiliations:** 1Department of Environmental Science and Engineering, Government College University Faisalabad, Faisalabad 38000, Pakistan; aroshamaqbool@gmail.com (A.M.); mrazi1532@yahoo.com (M.R.); msarif@outlook.com (M.S.A.); rida_akash@hotmail.com (T.Y.); mr548@ymail.com (M.R.); afzaalh345@gmail.com (A.H.); shamailanoureen@gcuf.edu.pk (S.N.); 2Department of Biological Sciences and Technology, China Medical University, Taichung 40402, Taiwan; 3Department of Environmental Sciences, The University of Lahore, Lahore 54000, Pakistan; 4Department of Zoology, College of Science, King Saud University, P.O. Box 2455, Riyadh 11451, Saudi Arabia; abdeldaim.m@vet.suez.edu.eg (M.M.A.-D.); salkahtani@ksu.edu.sa (S.A.); 5Pharmacology Department, Faculty of Veterinary Medicine, Suez Canal University, Ismailia 41522, Egypt

**Keywords:** N-fertilizers, toxicity, phytoremediation, antioxidants, metal uptake

## Abstract

Heavy metal contamination is currently a major environmental concern, as most agricultural land is being polluted from municipal discharge. Among various other pollutants, cadmium (Cd), one of the most harmful heavy metals, enters into the food chain through the irrigation of crops with an industrial effluent. In the present study, a pot experiment was designed to assess the effect of different nitrogen (N)-fertilizer forms in the phytoremediation of Cd through *Solanum nigrum* L. Two types of N fertilizers (NH_4_NO_3_ and urea) were applied to the soil in different ratios (0:0, 100:0, 0:100, and 50:50 of NH_4_NO_3_ and urea, individually) along with different Cd levels (0, 25, and 50 mg kg^−1^). The plants were harvested 70 days after sowing the seeds in pots. Cadmium contamination significantly inhibited the growth of leaves and roots of *S. nigrum* plants. Cadmium contamination also induced oxidative stress; however, the application of N-fertilizers increased the plant biomass by inhibiting oxidative stress and enhancing antioxidants’ enzymatic activities. The greatest plant growth was observed in the urea-treated plants compared with the NH_4_NO_3_-treated plants. In addition, urea-fed plants also accumulated higher Cd concentrations than NH_4_NO_3_-fed plants. It is concluded that urea is helpful for better growth of *S. nigrum* under Cd stress. Thus, an optimum concentration of N-fertilizers might be effective in the phytoremediation of heavy metals through *S. nigrum*.

## 1. Introduction

Anthropogenic activities have resulted in environmental degradation and a significant reduction in soil productivity [1]. Abiotic stress, such as heavy metals, extreme temperature, and salinity, affect crop production [2]. Industrial discharge, sewage sludge, mining, and agriculture have become major precursors of heavy metal discharge into water bodies and soil [3]. Compared with all other heavy metals, cadmium (Cd) is considered lethal once it becomes part of the food cycle.

Around 70% of the dietary intake of Cd occurs from consuming the vegetables and grain crops grown in Cd-contaminated soils [4]. Cadmium is highly toxic for plant growth and seed germination, in addition to the antioxidant, photosynthetic, and enzyme activities [5]. It also causes visual toxicity symptoms including a reduction in plant height, necrosis, and leaf chlorosis [6]. Excessive concentration of Cd also impairs the uptake of some integral nutrients, such as iron, zinc, and manganese [7]. Increased electrolyte leakage (EL), malondialdehyde (MDA) content, and oxidative stress in plants, along with a reduced enzyme activity, are also evident in Cd-affected plants [8]. Therefore, it is imperative to control Cd pollution so as to minimize the Cd concentrations in the soil, as well as to reduce its uptake by plants, especially in the edible portions of plants [9]. Phytoremediation is a useful technique to cope with the heavy metals problem, through use of green plants to remove/degrade pollutants from the contaminated soils. Phytoremediation is considered an attractive substitute for soil remediation, because it is one of the cheapest and most ecofriendly methods compared with other techniques [10].

Different plants accumulate heavy metals when grown in a metal contaminated soil; an example of such an uptake is the Cd taken up by *Brassica oleracea* L. [11]. In addition, various hyperaccumulator plants can help to remediate the highly metal-polluted soils. A number of Cd-hyperaccumulator plant species have been reported, including *S. nigrum* [12], *Malva rotundifolia* [13], *Bidens pilosa* [14], *Noccaea caerulescens* [15], *Sedum alfredii* [16], and *Jatropha curcas* [17]. Among these Cd hyperaccumulators, *S. nigrum* is considered more efficient for its fast growth and higher tolerance to Cd, without compromising biomass production [9]. However, the growth of *S. nigrum* may be negatively affected at higher Cd concentrations, which require some amendments, especially in the early growth stages, in order to improve the phytoextraction used by this plant.

The application of N-fertilizers is considered an effective measure to improve the soil fertility and phytoremediation efficiency of hyperaccumulator plants [18]. Nitrogen is an integral component of plants that makes various secondary metabolites, and is also helpful in chlorophyll, which is a key component of photosynthesis [19,20]. Numerous effects of N fertilization on soil–Cd dynamics have been observed, including desorption and adsorption, chemical transformation, dilution effect, and transportation, which ultimately influence Cd uptake by plants [21]. Many studies have shown that the addition of different N compounds (NH_4_ and NO_3_) in different combinations and dosages affect plant growth and biomass [22,23,24,25,26].

The study was designed to evaluate the effect of various N-fertilizers on the phytoremediation of Cd through *S. nigrum* by measuring changes in plant growth, chlorophyll and carotenoid contents, photosynthetic parameters, oxidative stress, and antioxidant enzyme activities under different N fertilizer combinations and Cd contamination levels.

## 2. Materials and Methods

### 2.1. Experimental Design

A pot experiment was conducted in a botanical garden located in Government College University Faisalabad (Faisalabad, Pakistan; 31°25′0″ N, 73°5′28″ E). The soil texture was measured by the method of Bouyoucos [27]; the pH and electrical conductivity (EC) with pH and EC meters, respectively; organic carbon was measured using the Walkley–Black method [28]; and the sodium absorption ratio (SAR) and soluble ions were measured using the method described by Page et al. [29]. The soil physicochemical characteristics are shown in Table 1. The soil was spiked and entirely mixed with different Cd (CdCl_2_ 2.5H_2_O) levels (0, 25, and 50 mg kg^−1^). Nitrogen fertilizers of two different kinds (NH_4_NO_3_ and urea) with varying ratios (0–0, 100–0, 0–100, and 50–50 mg kg^−1^) were added into the soil. It is noteworthy that the contents for Cd, Pb, Cu, and Zn were fundamentally low and unnoticeable in these fertilizers.

### 2.2. Soil Pot Experiments

In this study, 5% sodium hypochlorite (NaClO) was applied for 10 min in order to sterilize the *S. nigrum* seeds, which were then washed four times with deionized water. The washed and blot-dried seeds were planted in a plastic tray filled with sand, and half-strength Hoagland solution was applied. After three weeks of germination, the uniform seedlings were transferred into pots (four seedlings in each pot). Every single pot was filled with 5 kg of soil. All of the pots were set following complete randomized design (CRD) with four repeats for each treatment, and the water was topped up with tap water in order to maintain the 70% soil water-holding limit.

### 2.3. Plant Harvesting

The plants were harvested 70 days after sowing the seeds and were sectioned into shoots and roots. The length of the roots and shoots, numbers of leaves, leaf area, and fresh weight of plants were measured. The root and shoot samples were further dried in an oven for 72 h at 70 °C, and the dry weights were measured.

### 2.4. Determination of Photosynthetic, Chlorophyll, and Carotenoid Content Parameters

Fresh leaf samples (0.5 g) were soaked in acetone (85%, *v*/*v,* Sigma) and placed in the dark. The soaked samples were centrifuged (4000× *g* for 10 min, 4 °C) and the supernatant was collected. Data were recorded on a spectrophotometer at wavelengths of 470, 647, and 664.5 nm, separately. The chlorophyll content, i.e., chlorophyll a, chlorophyll b, and total chlorophyll, and carotenoids were recorded [30]. On a sunny day (10:00 a.m. to 12:00 p.m.), an infrared gas analyzer (IRGA) was used for the assessment of the conductance of the stomata water-use efficiency, rate of transpiration, and photosynthetic rate in the plants’ leaves.

### 2.5. Determination of EL, MDA, H_2_O_2_, and Antioxidant Enzyme Concentration

The measurement of both the oxidative stress markers and the activities of antioxidant enzymes were done 70 days after seed sowing. The shoot and root samples were placed in glass tubes vertically and heated at 32 °C for 2 h in distilled water of a known volume in order to measure the electrolyte leakage. This solution was termed EC_1_. Then, the same solution was heated at 121 °C for 20 min, and EC of this second solution was recorded and named EC_2_. The Dionisio-Sese and Tobita [31] equation was used to estimate the EL content.

For the estimation of the H_2_O_2_ content, a phosphate buffer solution (3.0 mL) was added to the sample (50 mg) and centrifuged at 6000× *g* for 30 min, keeping the temperature at 4 °C. Then, 1 mL of titanium sulfate (0.1%) was mixed in a supernatant and centrifuged at 6000× *g* for 20 min at 4 °C. Absorption was determined at a wavelength of 410 nm and a coefficient of extinction for H_2_O_2_ of 0.28 µmol^−1^ cm^−1^. The peroxidase (POD), catalase (CAT), superoxide dismutase (SOD), and ascorbate peroxidase (APX) enzyme activities were determined following the recommendations of Zhang [32] and Aebi [33]. The samples were prepared in a phosphate buffer (0.05 mmol) and the supernatant was gained through centrifuging for 10 min at 12,000× *g* and 4 °C.

### 2.6. Cadmium Determination in Plants, Translocation Factor and Bioaccumulation Factor

Crushed root and shoot samples were digested using HNO_3_-HClO_4_ (3:1, *v*/*v*). The samples were kept in 65% HNO_3_-HClO_4_ (3:1, *v*/*v*) for one night and then put on a hot plate after adding HNO_3_ (5.0 mL). The clear solution obtained after digestion was filtered and the Cd concentration was measured using an atomic absorption spectrophotometer. Translocation factor (TF) was calculated by using Equation (1):TF = Cd (Plant shoot)/Cd (Plant root)(1)
and the bioaccumulation factor (BCF) was calculated using Equation (2):TF = Cd concentration in root/Total Cd concentration in soil(2)

### 2.7. Statistical Analysis

A two-way analysis of variance test (ANOVA) was applied in order to test the significance of the N fertilizers and Cd concentrations. Tukey’s posthoc test was applied for the multiple means comparison technique. The statistical analysis was performed with SPSS for Windows Software v. 19 (IBM, Armonk, NY, USA).

## 3. Results

### 3.1. Assessment of N-Fertilizer on Growth and Biomass

Stunted growth was observed in the control group, while no such signs were observed in the N-fertilizer-treated group (Figure 1). The roots’ and shoots’ dry weight, the number of leaves per plant shoot length and root length, and the leaf area of the *S. nigrum* plants significantly increased in the N-fertilizer treatments (Figure 1). A maximum plant growth was observed in the urea-treated plants at a Cd level (0 mg kg^−1^), with respect to the NH_4_NO_3_-treated and control plants. In addition, more leaves were observed in the urea-treated plants compared with the NH_4_NO_3_-treated plants (Figure 1).

### 3.2. Assessment of Gas Exchange, Chlorophyll, and Carotenoid Content Attributes

Differential responses of chlorophyll and gas exchange attributes were recorded in urea-treated and NH_4_NO_3_-treated plants. Significant changes with the maximum values of Chl *a*, Chl *b*, total Chl, and carotenoid contents were observed in the urea-fed plants (Figure 2). A nonsignificant increase in chlorophyll content was observed in the urea-treated plants at 25 mg kg^−1^ Cd concentrations, in contrast to the NH_4_NO_3_-treated and control plants. In addition, a higher expression of gas attributes was recorded in the urea-treated plants than that of the NH_4_NO_3_-fed plants (Figure 3).

### 3.3. Assessment of Antioxidant Enzyme Activities and EL, MDA, and H_2_O_2_

After N-fertilizer supplementation, a significant reduction was noticed in the MDA, H_2_O_2_, and electrolyte leakage parameters (Figure 4). Reductions in EL in the leaves of the urea, NH_4_NO_3_, and urea + NH_4_NO_3_ treated plants were 15%, 33%, and 22%, respectively, compared with the control plants. The catalase (CAT), peroxidase (POD), ascorbate peroxidase (APX), and superoxide dismutase (SOD) enzyme activities in the *S. nigrum* leaf were significantly increased after the addition of N-fertilizers (Figure 5). In contrast to the control group, the increases in the POD, CAT, APX, and SOD values of the N-fertilized plants were 41%, 21%, and 64%; 47%, 22%, and 66%; 15%, 28%, and 42%; and 48%, 69%, and 25%, respectively.

### 3.4. Assessment of Concentration of Cd in Plants

The optimum induction of N-fertilizers remarkably enhanced the Cd concentration in the roots and shoots of *S. nigrum* compared with the respective controls (Figure 6). In order to assess the phytoremediation potential of Cd by *S. nigrum*, the TF and BCF were calculated. Results showed that plants supplemented with urea had higher BCF when compared with other treatments.

## 4. Discussion

### 4.1. Biomass and Plant Growth

It is an established fact that fertilizer application generally has positive effects on plant growth under heavy metal toxicity [34,35]. In the present study, the highest plant growth was observed in the urea-treated plants at 25 mg kg^−1^ Cd concentrations, in contrast to the NH_4_NO_3_-treated plants (Figure 1). The findings of our study are in accordance with the finding of Lin et al. [36], who found that 2.5 mmol N L^−1^ was the optimal concentration to boost *S. alfredii* shoot growth. Moreover, 1.0 mmol N L^−1^ was found to be an optimal concentration for Cd and Zn storage in the shoots of *S. alfredii*. Furthermore, stunted plant growth under heavy metal stress was noticed. For example, Rabelo et al. [37] reported that 2.0 mM Cd in *Tanzania guinea* grass inhibited the production of new tillers and leaves. A reduced nitrate uptake from the nutrient solution was reported by Gouia et al. [38] for Cd-exposed plants. It was investigated that the Cd concentration increased in *S. nigrum* with a supply of N-fertilizer (NH_4_NO_3_) without affecting the Cd speciation in plants [39]. Wei et al. [40] confirmed that the accretion of Cd in *S. nigrum* shoots and roots is augmented by applying urea in the growth medium. In our study, the root and shoot dry weights, shoot length, numbers of leaves per plant, and roots length of *S. nigrum* plants were significantly enhanced with the N-fertilizer application. In our current study, disparity in the Cd absorption in differently treated plants was observed, which is in line with the results of Ye et al. [41], who described an enhanced absorption of Cd and a phytoextraction efficiency of *T. patula* after N-fertilizer treatment. Moreover, our results are also supported by Yang et al. [42].

### 4.2. Photosynthetic Pigments

Maximum chlorophyll content was observed in the urea-treated plants at a 25 mg kg^−1^ Cd concentration, compared with the NH_4_NO_3_-treated and control plants (Figure 2). Our results support the previous reports [43], in which the author claimed that different N forms, e.g., ammonium nitrate (NH_4_NO_3_), ammonium sulfate ((NH_4_)_2_SO_4_), and calcium nitrate (Ca (NO_3_)_2_), significantly altered the Chl *a*, Chl *b*, total Chl, and carotenoid contents under Cd stress. Cadmium is a phytotoxic metal that causes growth inhibition. A high dose or prolonged exposure to Cd can lead to the death of plants, which is the result of disturbed respiration, reduced photosynthesis, and altered assimilation of N in plants [44]. Cadmium toxicity alters the N metabolism directly or indirectly [45]. It was found that an optimum N dosage may alleviate Cd toxicity to plants by improving the photosynthetic activity, stromal proteins, and the plant growth and biomass [46]. Under stress conditions, plants provoke antioxidant enzymes and certain metabolite activities for their survival in order to negate stress [47]. Jalloh et al. [48] reported that an appropriate dosage of urea and NH_4_^+^-N under Cd stress stimulated and increased the SOD and POD activities. Significantly increased MDA content in Cd-stressed plants were observed with addition of NH_4_^+^-N and/or urea in rice plants at the milking stage.

### 4.3. Oxidative Stress, Antioxidant Enzymes, and Cd Concentration

Malmir [49] found that Cr-induced oxidative stress, subsequently increasing the H_2_O_2_ and MDA contents. Similar evidence of Cr-induced oxidative stress was observed in *Helianthus annuus* L. [50] and *Brassica.* Various studies have shown that different amendments have reduced the Cd toxicity in different plant species, for example, EDTA reduced Cd toxicity in *Brassica* [51], silicon in *Brassica napus* L. [52], hydrogen peroxide in *Brassica napus* L. [53], supermine in *Vigna radiate* L. [54], and selenium in rapeseed seedlings [55]. The markedly reduced EL and MDA contents in the N-fertilizer-treated, Cd-stressed plants in our study demonstrate the significant role of N fertilizers for Cd stress. Our results revealed that SOD, POD, CAT, and APX activities of *S. nigrum* were recorded in response to Cd stress. Furthermore, the POD, CAT, SOD, and APX activities significantly increased after the application of N-fertilizers. Similar results were observed previously [43], indicating that N forms had a prominent impact on oxidative stress inflicted by Cr toxicity. Therefore, the Ca(NO_3_)_2_-fed plants had minimal oxidative stress compared with the (NH_4_)_2_SO_4_ and urea-treated plants.

Similar findings were observed for the N fertilizers for the phytoremediation of Cd, and even with different environmental conditions, soil textures, and experimental durations [56], and only a few analyses, we acknowledged the possible mechanisms for Cd phytoremediation with detailed analysis in our recent study. In addition, urea-treated plants translocated and accumulated a higher concentration of Cd than the NH_4_NO_3_-treated plants. However, the application of N-fertilizers inhibited oxidative damage and enhanced antioxidants’ enzyme activities. Our results depicted that at all levels of applied Cd, the both TF and BCF were enhanced with the application of urea alone, as compared with all other treatments (Figure 6E,F). This shows that among the various treatments, the plants treated with urea alone had greater potential for Cd phytoremediation. In conclusion, the suitable concentration of urea remarkably promoted and assisted the processes of Cd phytoremediation by *S. nigrum*.

## 5. Conclusions

Our study concludes that the application of an appropriate dosage of N fertilizers (NH_4_NO_3_ and urea) could be a suitable practice to enhance the remediation of heavy-metal-polluted soils when growing *S. nigrum*. The plant biomass increased significantly with N fertilizer addition. Maximum plant growth was observed in the urea-treated plants compared with the NH_4_NO_3_-treated plants. In addition, the urea-treated plants also accumulated a higher Cd than the NH_4_NO_3_-treated plants. The application of N fertilizers inhibited oxidative stress and enhanced the antioxidants’ enzymatic activities. It is concluded that urea might be helpful for better growth of *S. nigrum* under Cd stress. Our results indicate that the best N-fertilizer is urea, and that it is recommended for the phytoremediation of Cd. However, the application of urea for heavy metals phytoremediation other than Cd is still unknown, and future studies can be helpful for this investigation.

## Figures and Tables

**Figure 1 ijerph-17-03850-f001:**
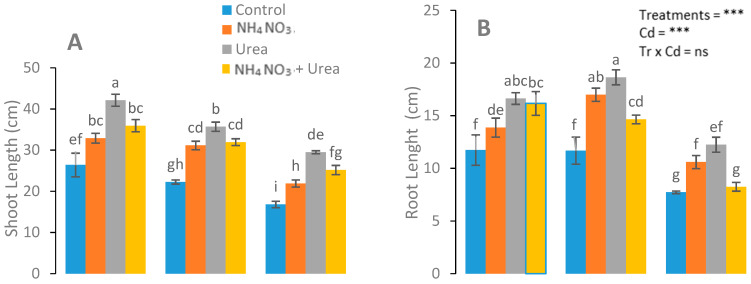
Effect of the Cd (0, 25, and 50 mg kg^−1^) and N fertilizers (ratios of 0:0, 100:0, 0:100, and 50:50 mg kg^−1^ urea, NH_4_NO_3_, and urea + NH_4_NO_3_) on shoot length (**A**), root length (**B**), shoot dry weight (**C**), number of leaves (**D**), root dry weight (**E**), and leaf area (**F**) of *S. nigrum*. Different letters show a significance difference at *p* ˂ 0.05 along with *n* = 4.

**Figure 2 ijerph-17-03850-f002:**
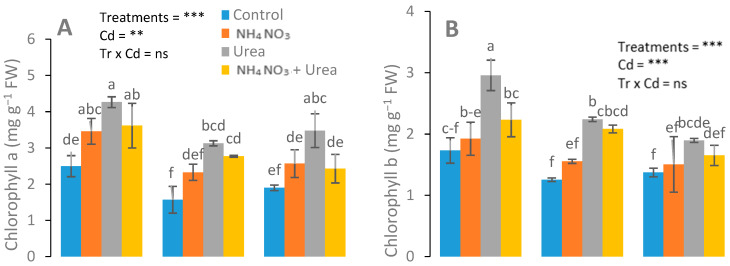
Effects of Cd (0, 25, and 50 mg kg^−1^) and N fertilizers (ratios of 0:0, 100:0, 0:100, and 50:50 mg kg^−1^ urea, NH_4_NO_3_, and urea + NH_4_NO_3_) on chlorophyll a (**A**), chlorophyll b (**B**), and carotenoid (**C**) of *S. nigrum*. Different letters show a significance difference at *p* ˂ 0.05 along with *n* = 4.

**Figure 3 ijerph-17-03850-f003:**
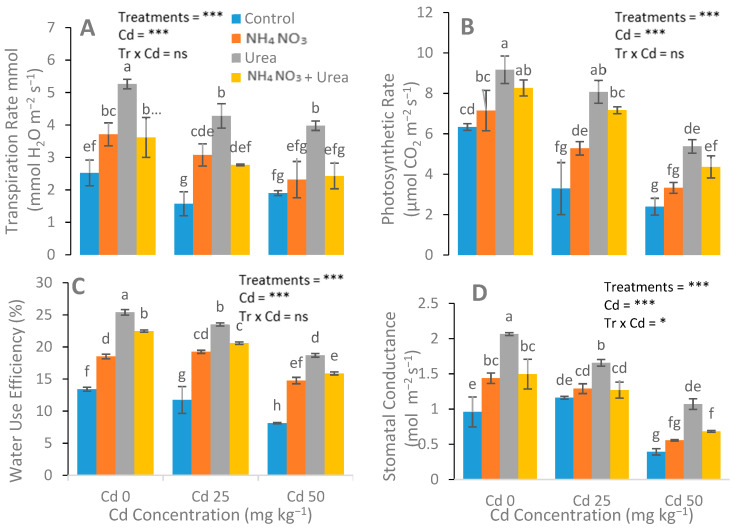
Effects of Cd stress (0, 25, and 50 mg kg^−1^) and N fertilizers (ratios of 0:0, 100:0, 0:100, and 50:50 mg kg^−1^ urea, NH_4_NO_3_, and urea + NH_4_NO_3_) on the transpiration rate (**A**), photosynthetic rate (**B**), water-use efficiency (**C**), and stomata conductance (**D**) of *S. nigrum*. Different letters show a significance difference at *p* ˂ 0.05 along with *n* = 4.

**Figure 4 ijerph-17-03850-f004:**
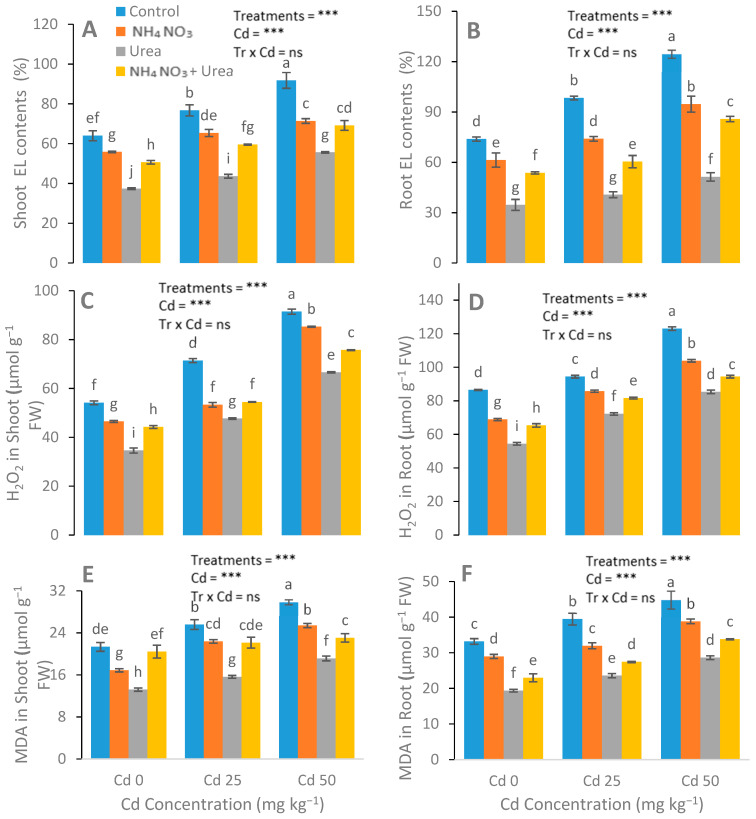
Effect of Cd (0, 25, and 50 mg kg^−1^) and N fertilizers (ratios of 0:0, 100:0, 0:100, and 50:50 mg kg^−1^ urea, NH_4_NO_3_, and urea + NH_4_NO_3_) on electrolyte leakage (EL) in leaves (**A**), EL in roots (**B**), H_2_O_2_ in leaves (**C**), H_2_O_2_ in roots (**D**), malondialdehyde (MDA) in leaves (**E**), and MDA in the roots (**F**) of *S. nigrum*. Different letters show a significance difference at *p* ˂ 0.05 along with *n* = 4.

**Figure 5 ijerph-17-03850-f005:**
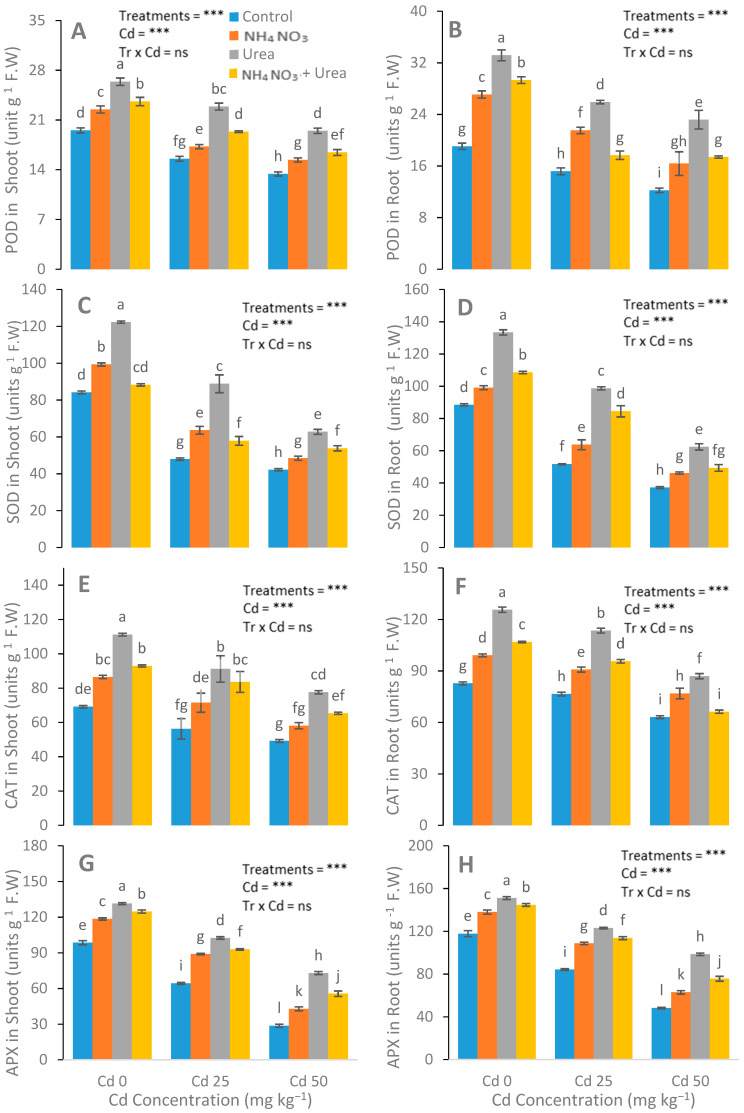
Effects of Cd (0, 25, and 50 mg kg^−1^) and N fertilizers (levels 0–0, 100–0, 0–100, and 50–50 mg kg^−1^ for urea, NH_4_NO_3_, and urea + NH_4_NO_3_) on POD in leaves (**A**), POD in roots (**B**), SOD in leaves (**C**), SOD in roots (**D**), CAT in leaves (**E**), CAT in roots (**F**), APX in leaves (**G**), and APX in roots (**H**) of *S. nigrum*. Different letters show a significance difference at *p* ˂ 0.05 along with *n* = 4.

**Figure 6 ijerph-17-03850-f006:**
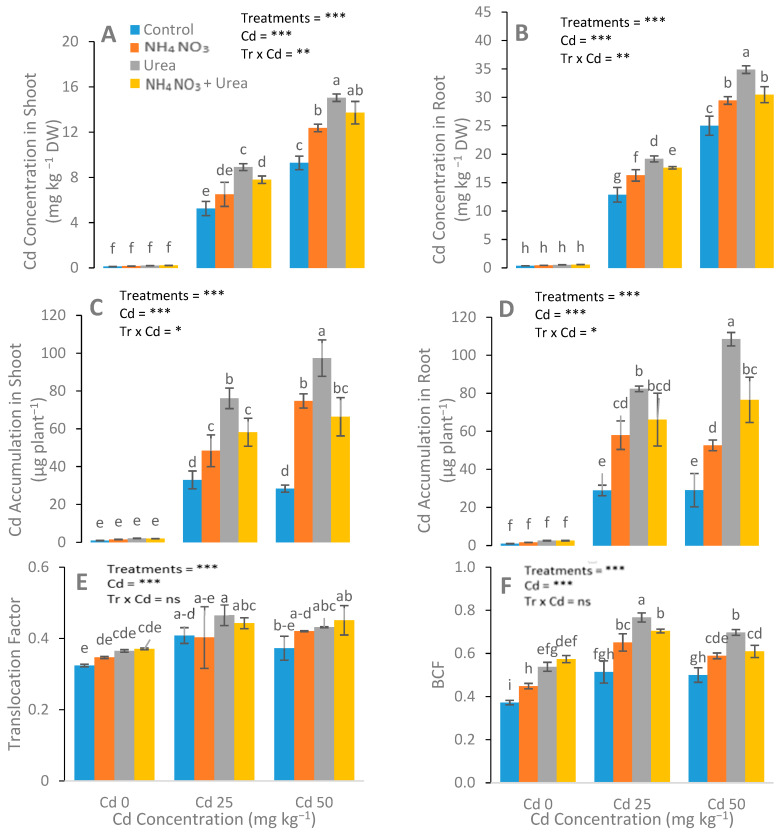
Effect of Cd stress (0, 25, and 50 mg kg^−1^) and N fertilizers (levels 0–0, 100–0, 0–100, and 50–50 mg kg^−1^ for urea, NH_4_NO_3_, and urea + NH_4_NO_3_) on Cd uptake in shoots (**A**), Cd uptake in roots (**B**), Cd accumulation in shoots (**C**), Cd accumulation in roots (**D**) translocation factor (**E**) and bioaccumulation factor (BCF) (**F)** of *S. nigrum*. Different letters show a significance difference at *p* ˂ 0.05 along with *n* = 4.

**Table 1 ijerph-17-03850-t001:** Soil physicochemical properties used for the experiment.

Texture	Sandy Loam
Silt	14.72%
Sand	68.07%
Clay	17.21%
EC	1.83 dS m^−1^
pH	7.72
SAR	1.93 (mmol L^−1^)^1/2^
Available P	2.20 mg kg^−1^
Organic matter	0.64%
HCO_3_^−^^1^	2.58 mmol L^−1^
SO_4_^−^^2^	11.69 mmol L^−1^
Cl^−^	5.35 mmol L^−1^
Ca^2+^ + Mg^2+^	14.26 mmol L^−1^
K^+^	0.03 mmol L^−1^
Na^+^	5.48 mmol L^−1^
Available Zn^2+^	0.81 mg kg^−1^
Available Cu^2+^	0.34 mg kg^−1^
Available Cd^2+^	0.09 mg kg^−1^

In table, EC stands for electrical conductivity and SAR for sodium absorption ratio.

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
