# Peer review of "N-Fertilizer (Urea) Enhances the Phytoextraction of Cadmium through Solanum nigrum L."

_ijerph, 2020, doi:10.3390/ijerph17113850_

Round 1

Reviewer 1 Report

The research presented by the authors is interesting and probably worthy of being published in this journal. Notwithstanding, there are many languages mistakes (a few of them are highlighted below) and Tables are not clear enough, at least from my point of view. My main suggestion is to calculate the translocation and bioaccumulation factors to assess the actual potential of Solanum nigrum for phytoremediation by comparing the results obtained with other plants by other authors.

Other points raised:

Figures. Subscripts of chemical formulas (NH4NO3) must be added. Besides, figures are hard to interpret. For example, "control" stands for "N fertilizer level 0-0", but readers have to guess it. Words are sometimes in capital letters and other not (for instance, the word "shoot" in the different figures). Figures should be improved.

Page 2, lines 14-17. “A number of Cd-hyperaccumulators plant species have been reported including Solanum nigrum [12], Malva rotundifolia [13], Bidens pilosa [14], Noccaea caerulescens [15] and Sedum alfredii [16]. Among these Cd hyperaccumulator, S. nigrum is considered more efficient for its fast growth and higher tolerance for Cd without compromising biomass production [9]. There are many other hypperaccumulator plants, such as Jatropha curcas, to remove Cd. To be specific, J. curcas has been reported to completely remove Cd from mining soils while increasing biomass growth (Phytoremediation of highly contaminated mining soils by Jatropha curcas L. and production of catalytic carbons from the generated biomass, Journal of Environmental Management, 2019, 231, 886-895). Mining soils mean soils with high concentrations of many heavy metals, not only Cd, which accounts for the potential of this plant for phytoremediation. Probably this part of the introduction section could be enhanced by adding more references about plants able to uptake Cd. These new references could be also used to enrich the Discussion section.

Page 2. line 37. S. nigrim plants? Do authors mean S. nigrum (in italic) plants?

Page 2, line 36, lines 39-40, etc. “… were notice …” Please check English grammar throughout the text.

Page 3, line 6. “attributes has been recorded” should be changed by “attributes have been recorded”.

Page 5, line 8, “respectively”. A comma is missing.

Page 5, line 1; page 8, line 4. S. nigrim???? Please correct.

Page 7, line 14 “It is an established fact that fertilizer application generally HAS”.

Page 7, line 17. Change “which found” by “who found”.

Page 8, line 11. "Our results support the previous reports [34] who claimed". Please rephrase this sentence since "who" is not pertinent.

Page 8, lines 39-40. "studies we have did the detailed analysis with describing the possible mechanisms for phytoremediation of Cd.". Please rephrase and correct language mistakes (a verb is also needed).

Page 8, lines 40-41. “In addition, urea-treated plants translocated and accumulated higher concentration of Cd than NH4NO3-treated plants”. To talk about translocation, it is need the concentration of Cd in soils. I think that, as stated at the beginning of my review, translocation and bioaccumulation factors should be calculated. From these results authors could claim whether S. nigrum is hyperaccumalador as well as its suitability for phytostabilization (phytosequestration) or phytoextraction (phytoaccumulation),

Material and Methods. How were soil physico-chemical properties of Table 1 measured?

Page 10, line 1. “As the estimation of H2O2, Phosphate buffer solution”. Please rephrase.

Page 10, lines 2 and 3 “The titanium sulfate (1.0 mL) 0.1% was mixed in supernatant”. Please rephrase and explain clearly.

Page 10, line 6. Change “(0.05M)” by “(0.05 M)”.

Page 10, line 19. “Our study conclude that application”. Please correct the language mistake.

Main conclusion. “Our study concludes that application of an appropriate dosage of N fertilizers (NH4NO3 and urea) could be a suitable practice in order to enhance remediation of the Cd polluted soils by growing S. nigrum.”. This conclusion is issued from an ideal soil in which there is only a metal (Cd) at relative high concentration (in fact, the concentrations assayed are not high at all. Real contaminated soils contain much higher concentrations of different pollutants. For instance, in the above mentioned reference there are metals at concentration higher than 3,000 mg kg-1. Therefore, I suggest the authors, if they are going to pursuing this research, to plant S. nigrum in, for example, mining soils (adding of not N-fertilizers), to assess what happens. Maybe the results obtained when there are only Cd do no occur when there are many other metals.

Author Response

Reviewer 1

The research presented by the authors is interesting and probably worthy of being published in this journal. Notwithstanding, there are many languages mistakes (a few of them are highlighted below) and Tables are not clear enough, at least from my point of view. My main suggestion is to calculate the translocation and bioaccumulation factors to assess the actual potential of Solanum nigrum for phytoremediation by comparing the results obtained with other plants by other authors.

Response

Thank you very much for your positive and valuable comments. We have tried our best to respond your comments and have improved our revised manuscript while considering your comments. We have improved the manuscript language by a native English speaker, the certificate for language improvement has been attached herewith. We have calculated the translocation and bioaccumulation factors as well as improved the study table in our revised version as per your nice suggestion.

Other points raised: Figures. Subscripts of chemical formulas (NH4NO3) must be added. Besides, figures are hard to interpret. For example, "control" stands for "N fertilizer level 0-0", but readers have to guess it. Words are sometimes in capital letters and other not (for instance, the word "shoot" in the different figures). Figures should be improved.

Response

Thanks for catching this. We have added the subscripts of said chemical formula and have made sure the words symmetry of figures in our revised manuscript.

.

Page 2, lines 14-17. “A number of Cd-hyperaccumulators plant species have been reported including Solanum nigrum [12], Malva rotundifolia [13], Bidens pilosa [14], Noccaea caerulescens [15] and Sedum alfredii [16]. Among these Cd hyperaccumulator, S. nigrum is considered more efficient for its fast growth and higher tolerance for Cd without compromising biomass production [9]. There are many other hypperaccumulator plants, such as Jatropha curcas, to remove Cd. To be specific, J. curcas has been reported to completely remove Cd from mining soils while increasing biomass growth (Phytoremediation of highly contaminated mining soils by Jatropha curcas L. and production of catalytic carbons from the generated biomass, Journal of Environmental Management, 2019, 231, 886-895). Mining soils mean soils with high concentrations of many heavy metals, not only Cd, which accounts for the potential of this plant for phytoremediation. Probably this part of the introduction section could be

Enhanced by adding more references about plants able to uptake Cd. These new references could be also used to enrich the Discussion section.

Response

Thanks for good suggestion, we have cited the new study to improve our manuscript. See reference number 17.

Page 2. line 37. S. nigrim plants? Do authors mean S. nigrum (in italic) plants?

Response

Yes it is S. nigrum and change has been made throughout the manuscript.

Page 2, line 36, lines 39-40, etc. “… were notice …” Please check English grammar throughout the text.

Response

Thanks for catching this, the change has been made and we have checked English grammar as per your nice suggestion.

Page 3, line 6. “Attributes has been recorded” should be changed by “attributes have been recorded”.

Response

Thanks for catching this. Attributes has been recorded, replaced by “attributes have been recorded”.

Page 5, line 8, “respectively”. A comma is missing.

Response

Comma has been added.

Page 5, line 10; page 8, line 4. S. nigrim???? Please correct.

Response

We have made the correction.

Page 7, line 14 “It is an established fact that fertilizer application generally HAS”.

Response

Thanks for nice comment. We have made correction.

Page 7, line 17. Change “which found” by “who found”.

Response

Thanks for catching this. Which found, replaced by “who found” as per your nice suggestion.

Page 8, line 11. "Our results support the previous reports [34] who claimed". Please rephrase this sentence since "who" is not pertinent.

Response

Thanks for nice comment. We have made the correction.

Page 8, lines 39-40. "studies we have did the detailed analysis with describing the possible mechanisms for phytoremediation of Cd.". Please rephrase and correct language mistakes (a verb is also needed).

Response

We have rephrase the said sentence.

Page 8, lines 40-41. “In addition, urea-treated plants translocated and accumulated higher concentration of Cd than NH4NO3-treated plants”. To talk about translocation, it is need the concentration of Cd in soils. I think that, as stated at the beginning of my review, translocation and bioaccumulation factors should be calculated. From these results authors could claim whether S. nigrum is hyperaccumalador as well as its suitability for phytostabilization (phytosequestration) or phytoextraction (phytoaccumulation),

Response

Thanks for critical comment. We have calculated the translocation and bioaccumulation factors and have added relevant graphs in our revised manuscript.

Material and Methods. How were soil physico-chemical properties of Table 1 measured?

Response

We have provided the protocols for soil physic-chemical properties in our revised manuscript.

Page 10, line 1. “As the estimation of H2O2, Phosphate buffer solution”. Please rephrase.

Response

The said sentence has been rephrased as suggested.

Page 10, lines 2 and 3 “The titanium sulfate (1.0 mL) 0.1% was mixed in supernatant”. Please rephrase and explain clearly.

Response

Thanks for nice comment. We have rephrased and explained the sentence more clearly.

Page 10, line 6. Change “(0.05M)” by “(0.05 M)”.

Response

Thanks for catching this. We have made the correction.

Page 10, line 19. “Our study conclude that application”. Please correct the language mistake.

Main conclusion. “Our study concludes that application of an appropriate dosage of N fertilizers (NH4NO3 and urea) could be a suitable practice in order to enhance remediation of the Cd polluted soils by growing S. nigrum.”. This conclusion is issued from an ideal soil in which there is only a metal (Cd) at relative high concentration (in fact, the concentrations assayed are not high at all. Real contaminated soils contain much higher concentrations of different pollutants. For instance, in the above mentioned reference there are metals at concentration higher than 3,000 mg kg-1. Therefore, I suggest the authors, if they are going to pursuing this research, to plant S. nigrum in, for example, mining soils (adding of not N-fertilizers), to assess what happens. Maybe the results obtained when there are only Cd do no occur when there are many other metals.

Response

Thanks for your critical comment. We have revised the conclusion by replacing Cd polluting soil by talking about the general heavy metals polluted soil.

Reviewer 2 Report

General comments:

- The work provides some valuable information for the management of nitrogen in the phytoremediation of Cd-contaminated sites. However, in its present form, it cannot be accepted for publication because of the many problems listed here. I hope the authors consider the following comments for a future submission.

- The manuscript is well structured, but it is not written in good English. For example, there is a problem with subject-verb agreement already in the title ("N-Fertilizers Enhances..."). Thus, I strongly suggest to hire a proofreading service in order to ensure a high-quality text.

- My biggest concern is the depth of the results and discussion. They are limited to general comparisons between urea and NH4NO3, or between control and fertilized-plants, and that was not the aim of this work. The discussion is almost entirely compromised for being based on a poor interpretation of the results.

Specific comments (page; line number: comment):

1; 2: Which N-fertilizer specifically? Based on the results, clearly it is urea.

1; 26: Do you mean 70 days after transplanting? Because the ‘Materials and Methods’ section states that “seeds were planted in a plastic tray filled with sand” (see 4.2. Soil pot experiments).

1; 27: Since the scientific name was already mentioned in full, you may henceforth abbreviate the genus name.

1; 32-34: This last sentence must be more specific. Indicate that the best N-fertilizer is urea, and that it is recommended for phytoremediation of Cd, not heavy metals in general. The same applies to the last sentence of the ‘Conclusions’ section.

1; 35: I recommend using keywords that are not already in the title.

2; 12: Authors should give preference to scientific instead of common names.

2; 29-30: I suggest to change “dry weights, chlorophyll contents, electrolyte leakage (EL), peroxidase (POD) and superoxide dismutase (SOD) activities” to “plant growth, chlorophyll and carotenoid contents, photosynthetic parameters, oxidative stress and antioxidant enzyme activities”. The aforementioned abbreviations must be described ahead.

2; 31-32: This specific objective was not assessed. Therefore, I suggest to remove it.

2; 35-36: Figure 1 does not show any information regarding chlorosis. Besides, authors should be more specific when presenting the results, indicating also the letters of each figure. For example: Figure 1A.

2; 37: Do you mean S. nigrum? Scientific names should be italicized. What about leaf area?

2; 38-39: Based on which graph? This statement is too vague and incorrect. Looking at Figure 1A, the maximum value was actually observed in urea-treated plants not exposed to Cd.

2; 39-40: Are you sure that urea-treated plants had 70% more leaves than NH4NO3-treated plants? This number does not seem right.

3; 3: Root dry weight (D) and number of leaves (E) were presented inverted in the figure, i.e. number of leaves (D) and root dry weight (E).

3; 5: Do not forget to mention carotenoids.

3; 8: There is no mention to total Chl in the ‘Materials and Methods’ section.

3; 8-10: At this Cd level, Chl a is virtually the same with urea or NH4NO3.

3; 10-11: Again, this number (70%) does not seem right. How did you calculate it?

5; 3: Water-use efficiency (B) and photosynthetic rate (C) were presented inverted in the figure, i.e. photosynthetic rate (B) and water-use efficiency (C).

5; 5-7: Do not forget to mention H2O2.

5; 7: I believe the right order is “NH4NO3, urea and urea+NH4NO3”, not “urea, NH4NO3 and urea+NH4NO3”.

5; 10: Do you mean S. nigrum? Please correct it.

5; 14: Were these contents determined in leaves or in the whole shoot (like shown in the y-axis)? Because they are obviously different, and it says “shoot” in the ‘Materials and Methods’ section. The same question applies to antioxidant enzyme activities.

9; 1: How did the authors set the Cd levels? Is there a reference?

9; 2: Do theses ratios mean mg of N kg-1? I hope so, because urea has much more nitrogen than NH4NO3.

9; 15: What about leaf area? Explain how you calculated it.

9; 17: Include “chlorophyll and carotenoid contents”.

9; 18: Dou you mean 0.5 g? Please specify it.

9; 23: I believe the control was also evaluated, not only the treated plants.

9; 26: Again. Do you mean 70 days after transplanting? And I believe you also used root samples. Please correct it.

10; 2: If you decide to use comma to separate groups of thousands, do it throughout all the text.

10; 6: Be consistent. If you used mmol L-1 in Table 1, why using M instead of mol L-1.

10; 15: I suggest to remove this excerpt “effects on plants biomass and biochemical variables”.

12; 30-32: This reference was not cited in the text.

Author Response

Comments and Suggestions for Authors

General comments:

- The work provides some valuable information for the management of nitrogen in the phytoremediation of Cd-contaminated sites. However, in its present form, it cannot be accepted for publication because of the many problems listed here. I hope the authors consider the following comments for a future submission.

- The manuscript is well structured, but it is not written in good English. For example, there is a problem with subject-verb agreement already in the title ("N-Fertilizers Enhances..."). Thus, I strongly suggest to hire a proofreading service in order to ensure a high-quality text.

- My biggest concern is the depth of the results and discussion. They are limited to general comparisons between urea and NH4NO3, or between control and fertilized-plants, and that was not the aim of this work. The discussion is almost entirely compromised for being based on a poor interpretation of the results.

Response

Thank you very much for your valuable comments. We have tried our best to respond your comments and have improved our revised manuscript while considering your nice comments.

We have improved the manuscript language by a native English speaker, the certificate for language improvement has been attached herewith. Also, we have improved the discussion section keeping in mind the better interpretation of results as per your nice suggestion.

Specific comments (page; line number: comment):

1; 2: Which N-fertilizer specifically? Based on the results, clearly it is urea.

Response

Thanks for nice comment. Yes, its urea, we have revised the title.

1; 26: Do you mean 70 days after transplanting? Because the ‘Materials and Methods’ section states that “seeds were planted in a plastic tray filled with sand” (see 4.2. Soil pot experiments).

Response

No, harvesting was made 70 days after the seeds sowing in sand. We have made clear in revised manuscript.

1; 27: Since the scientific name was already mentioned in full, you may henceforth abbreviate the genus name.

Response

We have made the change in whole manuscript.

1; 32-34: This last sentence must be more specific. Indicate that the best N-fertilizer is urea, and that it is recommended for phytoremediation of Cd, not heavy metals in general. The same applies to the last sentence of the ‘Conclusions’ section.

Response

Thanks for valuable comment. We have revised the said sentence more specifically along with clear future recommendation.

1; 35: I recommend using keywords that are not already in the title.

Response

We have revised keywords as suggested.

2; 12: Authors should give preference to scientific instead of common names.

Response

We have replaced common names with scientific names in whole manuscript.

2; 29-30: I suggest to change “dry weights, chlorophyll contents, electrolyte leakage (EL), peroxidase (POD) and superoxide dismutase (SOD) activities” to “plant growth, chlorophyll and carotenoid contents, photosynthetic parameters, oxidative stress and antioxidant enzyme activities”. The aforementioned abbreviations must be described ahead.

Response

The change has been done as suggested in revised manuscript.

2; 31-32: This specific objective was not assessed. Therefore, I suggest to remove it.

Response

Thanks for the suggestion. We have removed it.

2; 35-36: Figure 1 does not show any information regarding chlorosis. Besides, authors should be more specific when presenting the results, indicating also the letters of each figure. For example: Figure 1A.

Response

We have removed the chlorosis and tried to be more specific in our revised manuscript as per your nice suggestion.

2; 37: Do you mean S. nigrum? Scientific names should be italicized. What about leaf area?

Response

Thanks for catching this. We have italicized the scientific names in whole manuscript and have mentioned about leaf area.

2; 38-39: Based on which graph? This statement is too vague and incorrect. Looking at Figure 1A, the maximum value was actually observed in urea-treated plants not exposed to Cd.

Response

Yes you are correct, we have revised the sentence.

2; 39-40: Are you sure that urea-treated plants had 70% more leaves than NH4NO3-treated plants? This number does not seem right.

Response

It was typing error and we have made the correction.

3; 3: Root dry weight (D) and number of leaves (E) were presented inverted in the figure, i.e. number of leaves (D) and root dry weight (E).

Response

Thanks for catching this, we are made the correction.

3; 5: Do not forget to mention carotenoids.

We have mentioned in revised version.

3; 8: There is no mention to total Chl in the ‘Materials and Methods’ section.

Response

Thanks for catching this. We have mentioned it as suggested.

3; 8-10: At this Cd level, Chl a is virtually the same with urea or NH4NO3.

Response

Yes you are right, we have made the change as observed.

3; 10-11: Again, this number (70%) does not seem right. How did you calculate it?

Response

Actually we had some mistake in calculation, so we have removed this false number in revise version.

5; 3: Water-use efficiency (B) and photosynthetic rate (C) were presented inverted in the figure, i.e. photosynthetic rate (B) and water-use efficiency (C).

Response

We have made the correction.

5; 5-7: Do not forget to mention H2O2.

Response

We have mentioned as suggested.

5; 7: I believe the right order is “NH4NO3, urea and urea+NH4NO3”, not “urea, NH4NO3 and urea+NH4NO3”.

Response

May be you are right but this is the actual order which we observed from results of our study.

5; 10: Do you mean S. nigrum? Please correct it.

Response

Yes, we have made the correction.

5; 14: Were these contents determined in leaves or in the whole shoot (like shown in the y-axis)? Because they are obviously different, and it says “shoot” in the ‘Materials and Methods’ section. The same question applies to antioxidant enzyme activities.

Response

We have determined all said contents in plant leaves.

9; 1: How did the authors set the Cd levels? Is there a reference?

Response

From literature review, we set the Cd levels. A reference has been given. i.e.  Li, X.M.; Song, G.L. Cadmium uptake and root morphological changes in Medicago sativa under cadmium stress. Acta Prataculturae Sin. 2016, 25, 178-86.

9; 2: Do these ratios mean mg of N kg-1? I hope so, because urea has much more nitrogen than NH4NO3.

Response

 Yeah these are in mg kg-1

9; 15: What about leaf area? Explain how you calculated it.

Response

We have mentioned leaf area in revised version. Leaf area was measured through leaf area meter (L1 -2000, L1-COR, USA).

9; 17: Include “chlorophyll and carotenoid contents”.

Response

Thanks for catching this, we have mentioned in revised version.

9; 18: Dou you mean 0.5 g? Please specify it.

Response

Yes its 0.5 g, we have made the correction.

9; 23: I believe the control was also evaluated, not only the treated plants.

Response

Yes, all plants were evaluated. Correction has been made.

9; 26: Again. Do you mean 70 days after transplanting? And I believe you also used root samples. Please correct it.

Response

Plants were harvested after 70 days of seeds sowing and yes its shoot as well as root samples. We have clearly written these changes in revised manuscript.

10; 2: If you decide to use comma to separate groups of thousands, do it throughout all the text.

Response

Thanks for the suggestion. We have applied this rule throughout the manuscript where it needed.

10; 6: Be consistent. If you used mmol L-1 in Table 1, why using M instead of mol L-1.

Response

The change has been done as suggested in revised manuscript.

10; 15: I suggest to remove this excerpt “effects on plants biomass and biochemical variables”.

Response

Thanks for critical comment, the change has been made as per nice suggestion.

12; 30-32: This reference was not cited in the text.

Response

We have cited the said missing reference in our revised manuscript.

Round 2

Reviewer 1 Report

Authors have properly addressed most of my criticisms from the first round of review. However, I would have liked that authors had discussed the translocation and bioaccumulation factors that now they have calculated and included, but it is not mandatory. There are some minor style issues. For example, in figure 1, in axe x, Cd concentration is in the left in mg/kg and on the right in mg kg-1. In fact, along the text, tables and figures, concentrations are some times with / and others with -1. From the technical and scientific points of view, the paper is suitable for publication.

Author Response

Authors have properly addressed most of my criticisms from the first round of review. However, I would have liked that authors had discussed the translocation and bioaccumulation factors that now they have calculated and included, but it is not mandatory. There are some minor style issues. For example, in figure 1, in axe x, Cd concentration is in the left in mg/kg and on the right in mg kg-1. In fact, along the text, tables and figures, concentrations are some times with / and others with -1. From the technical and scientific points of view, the paper is suitable for publication.

Response

Thank you for your positive and appreciative comments. We have checked and made sure the similarity of said unit in whole manuscript. Once again, we thank the reviewer for spending his/her time on our manuscript to improve the manuscript under reviewer’s thoughtful comments.

Reviewer 2 Report

Authors have made some improvement in the manuscript. Therefore, I recommend publication.

Author Response

Authors have made some improvement in the manuscript. Therefore, I recommend publication.

Response

Thank you for your positive and appreciative comments. Again, we thank the reviewer for spending his/her time on our manuscript.